# ARDS after Pneumonectomy: How to Prevent It? Development of a Nomogram to Predict the Risk of ARDS after Pneumonectomy for Lung Cancer

**DOI:** 10.3390/cancers14246048

**Published:** 2022-12-08

**Authors:** Antonio Mazzella, Shehab Mohamed, Patrick Maisonneuve, Alessandro Borri, Monica Casiraghi, Luca Bertolaccini, Francesco Petrella, Giorgio Lo Iacono, Lorenzo Spaggiari

**Affiliations:** 1Division of Thoracic Surgery, European Institute of Oncology (IEO) IRCCS, 20141 Milan, Italy; 2Division of Epidemiology and Biostatistics, European Institute of Oncology (IEO) IRCCS, 20141 Milan, Italy; 3Department of Oncology and Hemato-Oncology, University of Milan, 20141 Milan, Italy

**Keywords:** pneumonectomy, ARDS, risk classification, nomogram, lung cancer

## Abstract

**Simple Summary:**

In the modern era, characterized by parenchymal-sparing procedures, in some cases pneumonectomy remains the only therapeutic approach to achieving oncological radicality. One of the most feared complications is undoubtedly respiratory failure and ARDS. Its cause after pneumonectomy is still unclear, and the study of risk factors is a subject of debate. In this paper, we evaluate the main risk factors for ARDS of a large cohort of patients and we classify them in four classes of growing risk in order to quantify their postoperative risk of ARDS and facilitate their global management.

**Abstract:**

(1) Background: The cause of ARDS after pneumonectomy is still unclear, and the study of risk factors is a subject of debate. (2) Methods: We reviewed a large panel of pre-, peri- and postoperative data of 211 patients who underwent pneumonectomy during the period 2014–2021. Univariable and multivariable logistic regression was used to quantify the association between preoperative parameters and the risk of developing ARDS, in addition to odds ratios and their respective 95% confidence intervals. A backward stepwise selection approach was used to limit the number of variables in the final multivariable model to significant independent predictors of ARDS. A nomogram was constructed based on the results of the final multivariable model, making it possible to estimate the probability of developing ARDS. Statistical significance was defined by a two-tailed *p*-value < 0.05. (3) Results: Out of 211 patients (13.3%), 28 developed ARDS. In the univariate analysis, increasing age, Charlson Comorbidity Index and ASA scores, DLCO < 75% predicted, preoperative C-reactive protein (CRP), lung perfusion and duration of surgery were associated with ARDS; a significant increase in ARDS was also observed with decreasing VO2max level. Multivariable analysis confirmed the role of ASA score, DLCO < 75% predicted, preoperative C-reactive protein and lung perfusion. Using the nomogram, we classified patients into four classes with rates of ARDS ranking from 2.0% to 34.0%. (4) Conclusions: Classification in four classes of growing risk allows a correct preoperative stratification of these patients in order to quantify the postoperative risk of ARDS and facilitate their global management.

## 1. Introduction

In the modern era, compared to the past, the rate of pneumonectomy has drastically decreased, thanks to parenchymal-sparing procedures (broncho-vascular sleeves), to improvement of medical, biological and immunotherapeutic treatments and to radiotherapy for advanced lung cancers. However, in some cases pneumonectomy remains the only therapeutic approach to achieve oncological radicality. This procedure is not without complications and is associated with the highest postoperative morbidity [1,2] and mortality rates, ranging from 5% to 9% [3,4,5,6], among pulmonary resections.

One of the most feared complications is undoubtedly respiratory failure. On one side of this dreaded event waterfall, we found post-pneumonectomy respiratory failure (ARF) and acute lung injury (ALI); these represent grave and devastating complications, necessitating invasive mechanical ventilation (IMV). The other side of the coin is represented by acute respiratory distress syndrome (ARDS). Patients developing post-pneumonectomy ARDS have a significantly increased mortality rate (20–50%) [5,6,7]. For the first time since 1994, the American-European Consensus Conference (AECC) provided a definition for ALI, defining it as a “syndrome of inflammation and increased permeability with pulmonary edema” associated with “several clinical—radiological alterations, not correlated to left atrial or pulmonary capillary hypertension” [7].

The definition of ARDS was also given by the AECC in 1994, then modified in 2012 [8], and finally adapted in 2016 [9]. It consisted of “respiratory failure during 1 week, linked to a known insult or new/worsening respiratory symptoms, associated with unilateral opacities on chest radiograph or CT, not related to cardiac dysfunction or volume overload”.

In light of these latter data, ARDS was then defined by the absence of hydrostatic or cardiogenic pulmonary edema, partial pressure of arterial oxygen and fraction of inspired oxygen (PaO2/FiO2) 300 mmHg or less and classified into three categories of severity: mild (200 mmHg < PaO2/FiO2 < 300 mmHg), moderate (100 mmHg < PaO2/FiO2 < 200 mmHg) and severe (PaO2/FiO2 < 100 mmHg).

Reviewing the different series reported in the literature, ARDS after lung resection occurred in 1% to 8% of patients with an ARDS-related mortality rate ranging from 30% to 80% [4,5,6,7,10,11,12,13,14,15,16,17,18,19]. 

From this simple aspect, it is easy to understand the fundamental role of the prevention/immediate treatment of this serious and potentially deadly condition.

The cause of ARDS in patients after pneumonectomy is still unclear, and the study of risk factors is a subject of debate. Thus far, few studies have tried to outline pre-, peri- and postoperative risk factors. Most of these studies included small cohorts of patients and are incomplete, with a large panel of investigated prognostic data. 

We reviewed our department database, including all patients who underwent pneumonectomy during the period 2014–2021.

Our aim was to evaluate the main risk factors for the development of ARDS and to construct a preoperative risk score, in order to prevent its insurgence. 

## 2. Materials and Methods

In accordance with the STROBE (Strengthening the Reporting of Observational Studies in Epidemiology) statement [20], we retrospectively reviewed single-center experience between 2014 and 2021. We retrospectively reviewed pre-, peri- and postoperative characteristics from the medical and surgical records (Table 1, Table 2 and Table 3) of 211 patients who underwent pneumonectomy. Written informed consent to undergo the procedure and for the use of clinical imaging data for scientific or educational purposes, or both, was obtained from all patients before the operation.

This study was conducted in accordance with the ethical principles of the declaration of Helsinki.

### 2.1. Preoperative General Status

Clinical and general preoperative characteristics are shown in Table 1. We particularly evaluated sex, age, weight and height, BMI (Body Mass Index) and body surface area [21], preoperative comorbidities, preoperative treatments (chemotherapy, immunotherapy or biologic therapy, radiotherapy or combination of both) and prolonged use of corticosteroids. In agreement with anesthesiologists, we considered preoperative ASA score and CCI (Charlson Comorbidity Index) [22].

### 2.2. Preoperative Functional and General Status

For all patients we routinely performed global spirometry for the evaluation of FEV1 (forced expiratory volume in the first second), DLCO (diffusion lung CO), DLCO/VA (diffusion lung CO)/alveolar ventilation) and perfusion pulmonary scintigraphy in order to evaluate PpoFEV1 (predicted postoperative FEV1) and PpoDLCO (predictive postoperative DLCO). In the last 4 years, our routine tests have included the cardio-pulmonary stress test for evaluating VO2max. We created a new variable for lung function impairment defined by the diagnosis of COPD or a reduced value of DLCO predicted < 75%, indicating the presence of emphysema or interstitial disease, based on a recent meta-analysis [23]. This suggested that the DLCO predicted might be an important measurement for COPD patients in terms of severity, exacerbation risk, mortality, emphysema domination and presence of pulmonary hypertension. 

### 2.3. Main Pulmonary Artery Diameter and Normalized Pulmonary Artery Diameter 

We measured the axial and sagittal diameter of the main pulmonary artery (PAD) and ascending aorta (Ao) at the level of the pulmonary artery bifurcation on preoperative CT scans; we also calculated the AP/Ao ratio. Pulmonary artery diameters were considered as crude (PAD) and normalized (nPAD) for body surface area. Some authors demonstrated that higher PAD-nPAD is an important independent predictor of postoperative respiratory failure, ARDS and mortality in patients undergoing pneumonectomy for lung cancer [24,25].

### 2.4. Indexes of Inflammatory Status

The inflammatory preoperative status of the patients was investigated by the analysis of different parameters: albumin, pre-albumin, CRP (C-reactive protein) and complete blood count (neutrophils, lymphocytes, platelets and hemoglobin). Starting from the blood count measurements, we calculated various indexes referring to inflammatory status [24]:-Platelets to Lymphocytes Ratio (PLR) and albumin multiplying lymphocytes, known as the Prognostic Nutritional Index (PNI);-HALP amalgamated index, which is measured as hemoglobin (g/L) × albumin (g/L) × lymphocyte (/L)/platelet (/L);-Serum Polymorpho-nuclear Neutrophil to Lymphocytes Ratio (NLR);-Systemic Immune-inflammation Index (SII): serum platelets × neutrophil/lymphocytes;-Advanced Lung Cancer Inflammation Index (ALI): serum albumin × BMI/NLR; BMI = weight (kg)/height (m)^2^.

### 2.5. Peri- and Postoperative Anesthesiologist Management

In our analysis, we evaluated other anesthesiology and respiratory parameters: average tidal volume during intervention (MVE), average PEEP (positive end-expiratory pressure), necessity for and quantity of blood transfusions during and immediately after surgery, intraoperative fluid balance and necessity for NIV during the postoperative period. 

Patients were routinely extubated in the operating or recovery room after the intervention, and then transferred to the thoracic department or intensive care unit. Whenever extubation was not possible for anesthesiology or respiratory issues, the double-lumen tube was substituted by a 1-lumen tube. When patients were transferred to the ICU, conventional mechanical ventilation was performed (tidal volume 8 mL/kg, PEEP 5 cm H_2_O).

### 2.6. Definition of ARDS

ARDS was defined according to the 2012 Berlin definition: -Acute onset within 7 days after surgery with ventilation setting for positive end-expiratory pressure (PEEP) of ≥5 cm H_2_O and bilateral lung infiltration, detected through chest x-ray: cannot be fully explained by effusion, lobar, lung collapse or nodules;-Absence of hydrostatic or cardiogenic pulmonary edema;-Partial pressure of arterial oxygen and fraction of inspired oxygen (PaO2/FiO2) 300 mmHg or less and classified in 3 categories of severity: mild (200 mmHg < PaO2/FiO2 < 300 mmHg), moderate (100 mmHg < PaO2/FiO2 < 200 mmHg) and severe (PaO2/FiO2 < 100 mmHg).

### 2.7. Statistical Methods

The association between patient characteristics and the development of ARDS following surgery was assessed using Fisher’s exact test for categorical variables and the Mantel–Haenszel test for trend for ordinal variables. Continuous variables such as blood values of lung function parameters were either categorized using normal-range cut-off values or dichotomized using the median value. A difference in the distribution of continuous variables was also assessed using the non-parametric test of the median.

Univariable and multivariable logistic regression was used to quantify the association between preoperative parameters and the risk of developing ARDS after surgery. Odds ratios (ORs) and their respective 95% confidence intervals (CI) were used to quantify the risk. Variables proved to be significantly associated with outcome at univariate analysis were entered in a multivariable model. A backward stepwise selection approach was used to limit the number of variables in the final multivariable model to significant independent predictors of ARDS.

A nomogram was constructed based on the results of the final multivariable model, making it possible to estimate the probability of developing ARDS. Statistical analysis was performed using SAS software version 9.4 (SAS Institute, Cary, NC, USA) and the R 3.4.4 packages rms and Hmisc. Statistical significance was defined by a two-tailed *p*-value < 0.05.

## 3. Results

A total of 211 patients underwent pneumonectomy for lung cancer at the IEO between 2014 and 2021 (62 f, 149 m). Of these, 13.3% (28 patients) developed ARDS during the postoperative period; 3.8% (8 patients) developed only lung atelectasis requiring bronchoscopy. The 30 day mortality was 3.3% (7 out of 211 patients); 30-day mortality in patients developing ARDS was 14.3% (4 patients out of 28).

Clinical, biologic, peri- and postoperative outcomes are described in Table 1, Table 2 and Table 3.

### 3.1. Preoperative and Surgical Treatments

A total of 109 patients (51.7%) were preoperatively treated by chemotherapy (103 patients) or chemotherapy and immunotherapy (3 patients) or a combination of chemotherapy and radiotherapy (3 patients). We performed 107 (50.7%) right and 104 (49.3%) left pneumonectomies. In 51 cases (24%), we performed an extended pneumonectomy due to the involvement of vascular mediastinum structures. In 9 (4.2%) cases, a tracheal sleeve pneumonectomy was performed, due to tumor positioning <2 cm away from the tracheal carina, assisted by extracorporeal membrane oxygenation (ECMO).

### 3.2. Postoperative Findings

Postoperative pathologic results were 82 (38.9%) squamous cell carcinomas, 96 (45.5%) adenocarcinomas, 5 (2.4%) large cell tumors, 2 (0.9%) typical carcinoids, 2 (0.9%) atypical carcinoids, 7 (3.3%) small cell lung cancers, 6 (2.8%) adenosquamous carcinomas, 5 (2.4%) metastases, 3 (1.4%) pleomorphic carcinomas, 1 (0.5%) sarcoma, 1 (0.5%) lymphoma and 1 (0.5%) adenoid-cystic carcinoma. 

### 3.3. Predicting Factors of ARDS

In the univariate analysis, increasing age, CCI and ASA score, COPD or DLCO < 75% predicted, preoperative CRP, lung perfusion and duration of surgery were associated with the development of ARDS (Table 1, Table 2 and Table 3). ASA score was correlated with CCI but proved to be a stronger predictor of ARDS when both variables were entered in the multivariable model. No significant association was observed for other parameters of interest such as aorta diameter (AoD), PAD, normalized PAD or PAD/AoD ratio. VO2 max was recorded for only 53 patients (25.1%), 6 of whom developed ARDS. Based on these limited data, a significant increase in ARDS was observed with decreasing Vo2max level (Mantel–Haenszel *p* = 0.03). The associations with increasing age and duration of surgery lost statistical significance in the multivariable analysis, and these two variables were also removed from the final model (Table 4). Patients with an ASA score of 3–4 had a 2.91-fold risk (95% CI 1.08–7.80) of developing ARDS. The risk was 5.62 (95% CI 1.72–18.4) for patients with COPD or reduced DLCO% predicted, 3.55 (95% CI 1.32–9.54) for patients with elevated preoperative CRP (>5mg/L) and 5.77 (95% CI 2.13–15.6) for patients with perfusion ≥40% in the operated lung (Table 4 and Table 5). 

The rate of ARDS increased significantly with the number of risk factors present before intervention (Table 3). Only 3 patients (2.4%) out of 126 patients with no more than one risk factor developed ARDS, compared to 20 (25.0%) out of 80 patients with two or three risk factors and all 5 (100%) patients with four risk factors (Table 5). 

A nomogram was constructed based on this final multivariable model (Figure 1). Points were attributed to the four individual variables (61 points for ASA score 3–4, 72 points for CRP > 5 mg/L, 99 points for the presence of COPD or reduced DLCO% predicted and 100 points if perfusion of the resected lung was ≥40%). After summing the points obtained for each single predictor, the nomogram allows the direct reading of the probability of ARDS.

The predicted probability of ARDS was calculated for each of the 211 patients. Based on the median value in the entire series (4.9%), 96 patients (45.5%) were classified at low risk and 115 (54.5%) at high risk. Only 2 patients (2.1%) developed ARDS in the low-risk group compared to 26 (22.6%) in the high-risk group (*p* < 0.0001) (Figure 1). The rate of ARDS reached 34% (17/50) in patients classified in the highest risk quartile. 

## 4. Discussion

ARDS is unquestionably one of the worst and most feared complications after pneumonectomy; it is well known that development of ARDS is closely linked to any direct or indirect pulmonary insult. Thus, on the surface of the alveolar endothelium, we witnessed an upregulation of inflammatory cytokines and an increase in the growth of reactive oxygen species (ROS) and of activated neutrophils with the initiation of the inflammation cascade; these events increase micro-vascular alveolar permeability, and this last aspect is probably responsible for postoperative pulmonary edema, representing the first stage of ALI and ARDS [13,16,18,19].

Everything that accompanies this represents a dramatic series of events leading to ARDS. Reviewing the literature, most of the studies focusing on ARDS and pneumonectomy included small cohorts of patients and often did not consider many pre- or postoperative characteristics.

In our analysis, we included data concerning clinical and inflammatory status of the patients, their metabolic, respiratory and functional status and their intraoperative and anesthesiologist management. 

In the univariate analysis, increasing age, CCI and ASA score, COPD or DLCO < 75% predicted, preoperative CRP, lung perfusion and duration of surgery were associated with the development of ARDS. In particular, patients over 65 years had a two-fold risk compared to those under the age of 60; patients over 75 years had even a triple risk of developing ARDS. Several authors agree that age > 65 years represents an important risk factor for post-pneumonectomy ARDS [12,16,17,18]. 

COPD and smoking represent additional substantial risk factors; on one hand, they are associated with an increased rate of perioperative complications with possible bleedings and subsequent peri- and postoperative transfusion; on the other hand, they favor the onset of postoperative infections and pneumonia, and these conditions promote an endothelium insult and the subsequent activation of the inflammation cascade [2,15,16]. In addition, the toxic action of smoke-related inhalation particles probably leads to a depletion of glutathione surfactant production and to an alteration in epithelial cell permeability with an increased vulnerability to infectious complications [15,25]. These patients often have low FEV1-DLCO/VA values and their respiratory function is compromised; the association between lung function reduction and ARDS is widely demonstrated [13,14,16,17,18], as well as the role of preoperative lung perfusion scintigraphy. Kim et al. [16] reported that a perfusion fraction level superior to 35% of the resected lung was related to higher occurrence of ARDS and early mortality. If the perfusion of the resected lung is already low, the rebound effect linked to vascular bed reduction and subsequent postoperative pulmonary hypertension in the other lung will be reduced. Our analysis strongly shows the relationship between ARDS and preoperative lung perfusion > 40%. In light of these data, another risk factor in the literature is represented by the right side [18,21]; the right lung is normally predominant in terms of perfusion and ventilation and postoperative pulmonary arterial pressure is higher after right than left pneumonectomy [22,23], although data in our analysis did not corroborate this thesis. 

Instead, we found a strong relationship between preoperative VO2 max, calculated by cardio-pulmonary stress test, and ARDS; however, we did not focus on this aspect and did not consider it in the multivariable analysis because this parameter was available only for 53 patients out of 211. Indeed, we only recently introduced the cardio-pulmonary stress test in our routine preoperative tests (since 2019); this factor will be further investigated in the future.

Another emerging aspect from our analysis is the inflammatory status of the patients. In particular, preoperative CRP serum levels >5 favor the development of post-pneumonectomy ARDS. CRP is a direct indicator of a patient’s inflammatory status and its increase in synthesis within hours after tissue injury or infection suggests that it contributes to host defense and is part of the innate immune response [26,27,28]; it has been demonstrated that high levels of CRP are associated with higher complication rates after lung surgery and more generally with a poor prognosis [29,30,31]. CRP level is often modified in oncologic patients, even if they do not show inflammatory symptoms. Changes in tumor-related inflammatory cells are strictly linked to the degree of inflammatory response to tumors; indeed, a higher inflammatory response often indicates a worse prognosis. On one hand, high levels of hemoglobin, albumin and lymphocytes may be positively correlated with prognosis; on the other hand, high levels of platelets may be associated with poor prognosis. In wider terms, inflammatory status impacts the quality of life of patients, their immune response to cancer and particularly the metabolism of lung cancer and of the host.

Concerning perioperative management, the most important elements emerging from our analysis are perioperative fluid balance and the duration of the intervention. Intraoperative fluid balance has been poorly investigated in the literature, but it could have a fundamental contribution in the endothelial damage preceding ARDS. The associations between acute lung injury (ALI) and excessive fluid intake have been demonstrated in various reports [12,14,17,32]. Licker and colleagues [17] suggest that the administration of large quantities of fluids in the first 24 h can favor the risk of ALI/ARDS in the 72 h after surgery. In particular, the authors demonstrated an odds ratio of 1.2 per increase of 500 mL of perioperative fluid administration. This is strongly in accordance with our analysis; intraoperative fluid balance represents an important and independent risk factor for ARDS in patients undergoing pneumonectomy (*p*: 0.0009) with an odds ratio of 1.5 per increase of 500 mL of perioperative fluid administration. The fluid surcharge, adding to the increase in pulmonary vascular resistance, to the reduction in lymphatic drainage after lung amputation [12,33] and to the increase in the blood flow (from two to six times) into the remaining lung, determines an excessive intravascular volume; these events can injure the capillary endothelium and increase the protein quantity in the interstitial and alveolar space [17,34,35]. This condition becomes more severe with longer surgical time, probably due to long-lasting surgical stress.

We did not find any relationship between perioperative IMV (invasive mechanic ventilation), tidal volume, plateau pressure and positive end-expiratory pressure (PEEP) and ARDS. Some studies [12,17,18] suggest that reduced tidal volume (<10 mL/kg), pressure-controlled ventilation and reduced PEEP during lung surgery limit peak alveolar pressures and ensure maximum alveolar recruitment. On the other hand, high tidal volume and increased PEEP determine important barotrauma on endothelium cells [17,19], stretch-activation of cation channels, subsequent upregulation of inflammatory cytokines, augmentation of oxygen-derived free radicals and activated neutrophils, and finally, an increase in alveolar permeability.

We constructed a nomogram that allows us to define four growing risk classes. Classification was made on the basis of preoperative parameters emerging from multivariable analysis (ASA, CRP, DLCO < 75 and lung perfusion) (Figure 1). We excluded the other significant parameter in the multivariable analysis (intraoperative fluid balance) because of the need to preoperatively assess the risk of ARDS. Likewise, we did not include VO2max because of its unavailability for most patients. Patients in the lowest class (group 1) present only a 2% probability of developing ARDS, compared to 15.3% in group 3 and 34% in group 4 (*p* < 0.0001) (Figure 1). This nomogram, however, needs to be validated in an independent cohort prior being used in the clinic.

Our classification is easy, reliable and exclusively based on preoperative and easily available data; it allows a correct preoperative stratification of patients based on their functional (lung perfusion and DLCO <75%) and inflammatory (CRP level) status and their medical history (ASA score) in order to ensure the choice of the best treatment (between surgery and a more conservative treatment such as radiotherapy) for these patients and facilitate their global management. Thus, a correct interaction between thoracic surgeons, respiratory physicians, anesthesiologists, physiotherapists and dedicated ICU nurses is mandatory. The first step is the correct analysis of medical history and of lung function. The second step, especially for patients in the highest classes, if the treatment remains surgery, is the correct surgical and anesthesiologist management.

This study presents some limits. First, it is a retrospective study investigating about 8 years of experience. Secondly, the number of events is relatively small, and it does not allow us to reach a definitive conclusion. Data relating to some parameters such as DLCO and fluid balance were partially missing (about 10%); this aspect might have affected the results of the analyses. It could be interesting to create a homogeneous cohort from different specialized centers worldwide to obtain a larger sample of patients.

## 5. Conclusions

Classification of patients who underwent pneumonectomy for lung cancer in four growing risk classes allows a correct preoperative stratification based on their functional and inflammatory status and their medical history in order to ensure the best care for these patients and facilitate their global management. A multicenter validation cohort is needed in order to assess how the nomogram works outside the calibration cohort. 

## Figures and Tables

**Figure 1 cancers-14-06048-f001:**
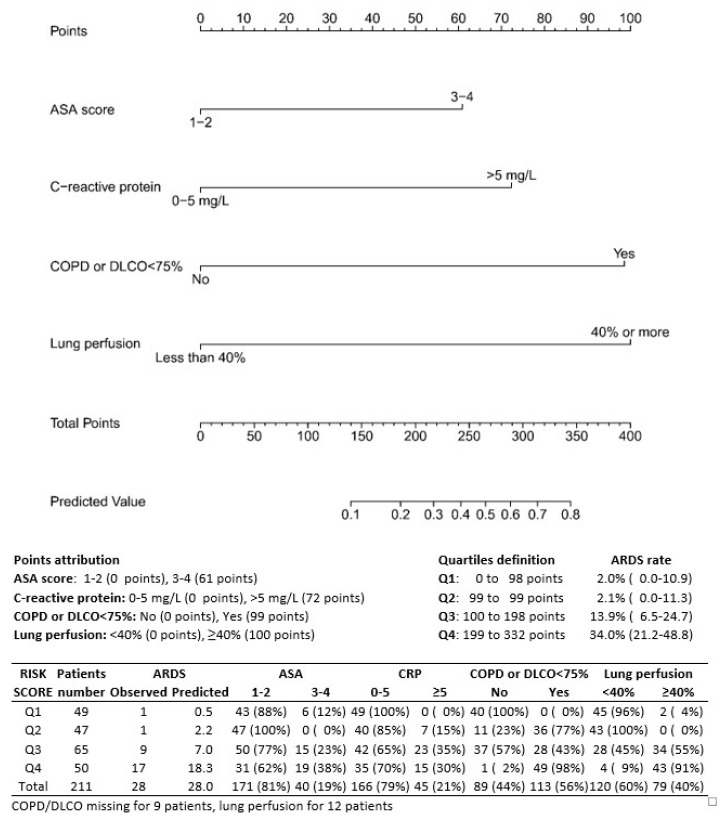
Nomogram for the prediction of ARDS; points attribution to different characteristics and division of the patients in 4 growing risk classes.

**Table 1 cancers-14-06048-t001:** Clinical factors associated with development of ARDS.

Patient Characteristics	Total	NO ARDS	YES ARDS	
	Patients (%)	Patients (%)	Patients (%)	*p*-Value
**Total**	211 (100.0)	183 (100.0)	28 (100.0)	
Age				
<60	61 (28.9)	57 (31.1)	4 (14.3)	
60–64	35 (16.6)	30 (16.4)	5 (17.9)	
65–69	60 (28.4)	51 (27.9)	9 (32.1)	
70–74	36 (17.1)	31 (16.9)	5 (17.9)	
75+	19 (9.0)	14 (7.7)	5 (17.9)	0.047
Sex				
Men	149 (70.6)	130 (71.0)	19 (67.9)	
Women	62 (29.4)	53 (29.0)	9 (32.1)	0.82
BMI				
Underweight	8 (3.8)	8 (4.4)	0 (0.0)	
Normal	100 (47.4)	83 (45.4)	17 (60.7)	
Overweight	85 (40.3)	75 (41.0)	10 (35.7)	
Obese	18 (8.5)	17 (9.3)	1 (3.6)	0.30
Area body surface				
Q1	70 (33.2)	60 (32.8)	10 (35.7)	
Q2	71 (33.6)	61 (33.3)	10 (35.7)	
Q3	70 (33.2)	62 (33.9)	8 (28.6)	0.90
Comorbidities				
Cardiac	62 (29.4)	51 (27.9)	11 (39.3)	0.27
Hypertension	110 (52.1)	92 (50.3)	18 (64.3)	0.22
Pulmonary	15 (7.1)	13 (7.1)	2 (7.1)	1.00
COPD	61 (28.9)	49 (26.8)	13 (46.4)	0.04
Cancer	23 (10.9)	18 (9.8)	5 (17.9)	0.20
Diabetes	20 (9.5)	16 (8.7)	4 (14.3)	0.31
Smoking				
Non-smoker	66 (31.3)	57 (31.1)	9 (32.1)	
Current smoker	40 (19.0)	32 (17.5)	8 (28.6)	
Ex-smoker	102 (48.3)	92 (50.3)	10 (35.7)	0.26
Alcohol				
No	207 (98.1)	180 (98.4)	27 (96.4)	
Yes	4 (1.9)	3 (1.6)	1 (3.6)	0.44
Presurgical treatment				
No	101 (47.9)	90 (49.2)	11 (39.3)	
Yes	110 (52.1)	93 (50.8)	17 (60.7)	0.42

**Table 2 cancers-14-06048-t002:** Functional, morphometric and biologic factors associated with development of ARDS.

Patient Characteristics	Total	NO ARDS	YES ARDS	
	Patients (%)	Patients (%)	Patients (%)	*p*-Value
**LUNG FUNCTION**				
FEV1 PPO (missing for 10)				
Median (range)	52 (25–131)	52 (27–131)	48 (25–96)	0.04
<50%	100 (49.8)	81 (47.1)	19 (67.9)	
≥50%	101 (50.2)	91 (52.9)	9 (32.1)	0.04
DLCO/VA (missing for 19 patients)				
Median (range)	51 (17–122)	52 (17–122)	48 (27–102)	0.30
<50%	96 (50.0)	80 (48.5)	16 (59.3)	
≥50%	96 (50.0)	85 (51.5)	11 (40.7)	0.41
DLCO PPO (missing for 13)				
Reduction (<75%)	75 (37.9)	60 (35.1)	15 (55.6)	
Normal (>75%)	123 (62.1)	111 (64.9)	12 (44.4)	0.05
COPD or DLCO < 75% (missing for 9)				
No	89 (44.1)	85 (48.9)	4 (14.3)	
Yes	113 (55.9)	89 (51.1)	24 (85.7)	0.0008
Lung perfusion (missing for 12)				
Median (range)	37 (2–54)	36 (3–51)	43 (2–54)	0.01
<40%	120 (60.3)	112 (64.7)	8 (30.8)	
≥40%	79 (39.7)	61 (35.3)	18 (69.2)	0.002
Vo2MAX				
Median (range)	19.7 (11.8–32.7)	19.9 (11.8–32.7)	17.0 (12.7–21.2)	0.10
Vo2MAX (missing for 158)				
T1 (<18)	17 (48.6)	13 (44.8)	4 (66.7)	
T2 (18.0–21.7)	18 (51.4)	16 (55.2)	2 (33.3)	
T3 (>21.7)	18 (51.4)	18 (62.1)	0 (0.0)	0.03
**VESSELS (missing for 6)**				
Aorta diameter				
≤median (32.1)	102 (49.8)	90 (50.8)	12 (42.9)	
>median (32.1)	103 (50.2)	87 (49.2)	16 (57.1)	0.54
Pulmonary artery diameter				
≤median (26.0)	103 (50.2)	89 (50.3)	14 (50.0)	
>median (26.0)	102 (49.8)	88 (49.7)	14 (50.0)	1.00
Normalized PAD				
≤median (9.80)	103 (50.2)	90 (50.8)	13 (46.4)	
>median (9.80)	102 (49.8)	87 (49.2)	15 (53.6)	0.84
PAD/AoD ratio				
≤median (0.8125)	103 (50.2)	89 (50.3)	14 (50.0)	
>median (0.8125)	102 (49.8)	88 (49.7)	14 (50.0)	1.00
**BLOOD PARAMETERS**				
White blood cells				
Low (<3.9)	2 (0.9)	2 (1.1)	0 (0.0)	
Normal (3.9–10.2)	153 (72.5)	134 (73.2)	19 (67.9)	
High (>10.2)	56 (26.5)	47 (25.7)	9 (32.1)	0.62
C-reactive protein				
Normal (≤5.0)	166 (78.7)	150 (82.0)	16 (57.1)	
High (>5.0)	45 (21.3)	33 (18.0)	12 (42.9)	0.006
Hemoglobin				
Low (<13.5)	140 (66.4)	119 (65.0)	21 (75.0)	
Normal (13.5–17.2)	69 (32.7)	62 (33.9)	7 (25.0)	
High (17.2)	2 (0.9)	2 (1.1)	0 (0.0)	0.54
Lymphocytes				
Low (<1.1)	24 (11.4)	21 (11.5)	3 (10.7)	
Normal (1.1–4.5)	183 (86.7)	158 (86.3)	25 (89.3)	
High (>4.5)	4 (1.9)	4 (2.2)	0 (0.0)	1.00
Neutrophils				
Low (<1.5)	2 (0.9)	2 (1.1)	0 (0.0)	
Normal (1.5–7.7)	167 (79.1)	146 (79.8)	21 (75.0)	
High (>7.7)	42 (19.9)	35 (19.1)	7 (25.0)	0.59
Platelets				
Low (<140)	5 (2.4)	4 (2.2)	1 (3.6)	
Normal (140–450)	186 (88.2)	164 (89.6)	22 (78.6)	
High (>450)	20 (9.5)	15 (8.2)	5 (17.9)	0.17
Albumin				
Low (<3.4)	26 (12.4)	20 (11.0)	6 (21.4)	
Normal (3.4–5.4)	184 (87.6)	162 (89.0)	22 (78.6)	0.13
**INFLAMMATORY SCORES**				
HALP score				
<0.35	106 (50.2)	89 (48.6)	17 (60.7)	
≥0.35	105 (49.8)	94 (51.4)	11 (39.3)	0.31
NLR score				
<3.0	114 (54.0)	101 (55.2)	13 (46.4)	
≥3.0	97 (46.0)	82 (44.8)	15 (53.6)	0.42
PLR score				
<150	106 (50.2)	94 (51.4)	12 (42.9)	
≥150	105 (49.8)	89 (48.6)	16 (57.1)	0.42
ALI index				
<35	107 (50.7)	89 (48.6)	18 (64.3)	
≥35	104 (49.3)	94 (51.4)	10 (35.7)	0.16
SII score				
<750	107 (50.7)	95 (51.9)	12 (42.9)	
≥750	104 (49.3)	88 (48.1)	16 (57.1)	0.42

FEV1: forced expiratory volume in the first second, DLCO: diffusion lung CO, COPD: chronic obstructive pulmonary disease, HALP: hemoglobin, albumin, lymphocyte and platelet score, NLR: Serum Polymorpho-nuclear Neutrophil to Lymphocytes Ratio, PLR: Platelets to Lymphocytes Ratio, ALI: Advanced Lung Cancer Inflammation Index (serum albumin × BMI/NLR; BMI = weight (kg)/height (m)^2^), SII: Systemic Immune-inflammation Index (serum platelets × neutrophil/lymphocytes).

**Table 3 cancers-14-06048-t003:** Peri- and postoperative factors associated with development of ARDS.

Patient Characteristics	Total	NO ARDS	YES ARDS	
	Patients (%)	Patients (%)	Patients (%)	*p*-Value
**SURGERY**				
Duration of surgery				
<180 min	117 (55.7)	107 (58.5)	10 (37.0)	
≥180 min	93 (44.3)	76 (41.5)	17 (63.0)	0.04
Side				
Right	107 (50.7)	89 (48.6)	18 (64.3)	
Left	104 (49.3)	94 (51.4)	10 (35.7)	0.16
Histology				
Adenocarcinoma	96 (45.5)	84 (45.9)	12 (42.9)	
Squamous or adenosquamous	88 (41.7)	72 (39.3)	16 (57.1)	
Other	27 (12.8)	27 (14.8)	0 (0.0)	0.03
Charlson Comorbidity Index (CCI)				
1–3	78 (37.0)	74 (40.4)	4 (14.3)	
4	69 (32.7)	60 (32.8)	9 (32.1)	
5–13	64 (30.3)	49 (26.8)	15 (53.6)	0.002
ASA				
1–2	171 (81.0)	155 (84.7)	16 (57.1)	
3–5	40 (19.0)	28 (15.3)	12 (42.9)	0.002
**ANESTH/VENTILATION**				
PEEP				
Median (range)	5 (0-12)	5 (0-12)	5 (3-10)	0.82
Postop NIV				
No	182 (86.3)	178 (97.3)	4 (14.3)	
Yes	29 (13.7)	5 (2.7)	24 (85.7)	<0.0001
Intraop ECMO				
No	203 (96.2)	178 (97.3)	25 (89.3)	
Yes	8 (3.8)	5 (2.7)	3 (10.7)	0.07
MVE				
Median (range)	8 (0.4–15)	8 (3–15)	8 (0.4–10)	0.95
Fluid balance (missing for 23)				
Median (range)	1100 (0–7700)	1036 (0–7700)	1468 (149–4900)	0.03
≤500	28 (14.9)	26 (15.9)	2 (8.3)	
>500–1000	54 (28.7)	49 (29.9)	5 (20.8)	
>1000–1500	55 (29.3)	49 (29.9)	6 (25.0)	
>1500	51 (27.1)	40 (24.4)	11 (45.8)	0.04
Corticosteroids (missing for 6)				
No	205 (97.2)	178 (97.3)	27 (96.4)	
Yes	6 (2.8)	5 (2.7)	1 (3.6)	0.34
Blood transfusion (PRE)				
No	187 (89.0)	161 (88.5)	26 (92.9)	
Yes	23 (11.0)	21 (11.5)	2 (7.1)	0.75
Blood transfusion (POST)				
No	162 (77.1)	143 (78.6)	19 (67.9)	
Yes	48 (22.9)	39 (21.4)	9 (32.1)	0.23
**HOSPITALIZATION**				
ICU				
Median (range)	1 (0–89)	1 (0–66)	1 (0–89)	0.0001
Hospital stay				
Median (range)	8 (0–180)	8 (0–180)	15 (7–180)	<0.0001
0–7 days	83 (40.9)	80 (44.2)	3 (13.6)	
7–14 days	87 (42.9)	80 (44.2)	7 (31.8)	
>14 days	33 (16.3)	21 (11.6)	12 (54.5)	<0.0001

ICU: intensive care unit, ECMO: extracorporeal membrane oxygenation, PEEP: positive end-expiratory pressure, NIV: noninvasive positive pressure ventilation.

**Table 4 cancers-14-06048-t004:** Preoperative (model 1) plus perioperative (model 2) predictors of ARDS in univariate and multivariable analyses.

Patient Characteristics	Univariate		Multivariable Model 1		Multivariable Model 2	
	OR (95% CI)	*p*-Value	OR (95% CI)	*p*-Value	OR (95% CI)	*p*-Value
ASA score						
1–2	1.00		1.00		1.00	
3–5	4.15 (1.78–9.71)	0.001	2.91 (1.08–7.80)	0.03	2.49 (0.88–7.11)	0.08
DLCO < 75% predicted						
No	1.00		1.00		1.00	
Yes	5.73 (1.91–17.2)	0.002	5.62 (1.72–18.4)	0.004	6.57 (1.84–23.4)	0.004
C-reactive protein						
Normal (≤5.0)	1.00		1.00		1.00	
High (>5.0)	3.41 (1.48–7.88)	0.004	3.55 (1.32–9.54)	0.01	3.85 (1.36–10.9)	0.01
Perfusion						
<40%	1.00		1.00		1.00	
≥40%	4.13 (1.70–10.1)	0.002	5.77 (2.13–15.6)	0.0006	8.32 (2.70–25.7)	0.0002
Intraoperative fluids						
per 500 mL increase	1.22 (1.01–1.46)	0.04			1.53 (1.11–2.11)	0.009

Odds ratio (OR) and 95% confidence intervals (CI) obtained from univariable and multivariable logistic regression. Only factors which retain statistical significance (*p* < 0.10) were included in the final multivariable model.

**Table 5 cancers-14-06048-t005:** Rate of ARDS and days in ICU and hospital according to number of risk factors for ARDS.

Number of Risk Factors *	Patients N (%)	ARDS N (%)	ICU	Hospital Stay
0–1 risk factors	126 (59.7)	3 (2.4)	1.2 ± 5.9; 1 [0–66]	10.5 ± 16.3; 8 [8–180]
2–3 risk factors	80 (37.9)	20 (25.0)	2.4 ± 10.4; 1 [0–89]	15.2 ± 21.9; 9 [5–180]
4 risk factors	5 (2.4)	5 (100)	4.2 ± 6.1; 2 [1–15]	11.0 ± 4.6; 11 [7–15]
*p*-Value		<0.0001 ^†^	0.001 ^‡^	<0.001 ^‡^

* ASA score (3–4), CRP > 5 mg/L, COPD or DLCO < 75 and lung perfusion >40%. ICU and hospital stay are expressed as mean ± standard deviation; median (range). ^†^ Mantel–Haenszel test for trend; ^‡^ *p* based on Spearman correlation.

## Data Availability

The original contributions presented in the study are included in the article.

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
