# Peer review of "ARDS after Pneumonectomy: How to Prevent It? Development of a Nomogram to Predict the Risk of ARDS after Pneumonectomy for Lung Cancer"

_cancers, 2022, doi:10.3390/cancers14246048_

Round 1
Reviewer 1 Report
I thank you for giving me the opportunity to review the manuscript entitled “ARDS after pneumonectomy: how to prevent it? Development of a nomogram to predict the risk of ARDS after pneumonectomy for lung cancer” by Mazzella A et al. The authors investigated 211 patients who underwent pneumonectomy for NSCLC, and 23 out of the 211 patients (13.3%) postoperatively developed ARDS. Multivariable analysis revealed that ASA score, DLCO<75% predicted, preoperative C-reactive protein, and lung perfusion as risk factors for postoperative ARDS in those patients.
This study focused on an interesting topic for general thoracic surgeons. The authors evaluated a lot of pre- and perioperative parameters, and the four risk factors identified by the multivariate analysis seem reasonable. Readers of the journal may have an interest in the results of this study, and the nomogram may help predict an incidence of postoperative ARDS in NSCLC patients who undergo pneumonectomy.
However, the result of the validation analysis might not have any meaning because the performance of the nomogram should be effective on the same data set. The performance of the nomogram might have been assessed on the different data sets.
1) All abbreviations used in the manuscript should be explained at the first mention, such as BMI, ASA score, CCI, FEV1, and DLCO.
2) Postoperative ARDS should be defined in the Material and Methods section.
3) When looking at the tables, some of the patients had missing data in many of the variables. How did you deal with such patients with missing data in the analyses?
4) Even in the variables regarding vessels, the data of such variables seem unknown in some patients. Was not preoperative evaluation by CT of the thorax performed in every patient?
5) As I mentioned above, the result of the validation analysis might not have any meaning because the performance of the nomogram should be effective on the same data set. The performance of the nomogram might have been assessed on the different data sets.
6) In the tables, all abbreviations used should be explained, such as FEV1, DLCO, and BPCO.
7) It is easier to understand if the columns of % are present in the patients with and without ARDS groups, respectively.
8) In the tables, the authors should present the number of patients with unknown data in every variable.
Author Response
REVIEWER 1
I thank you for giving me the opportunity to review the manuscript entitled “ARDS after pneumonectomy: how to prevent it? Development of a nomogram to predict the risk of ARDS after pneumonectomy for lung cancer” by Mazzella A et al. The authors investigated 211 patients who underwent pneumonectomy for NSCLC, and 23 out of the 211 patients (13.3%) postoperatively developed ARDS. Multivariable analysis revealed that ASA score, DLCO<75% predicted, preoperative C-reactive protein, and lung perfusion as risk factors for postoperative ARDS in those patients.
This study focused on an interesting topic for general thoracic surgeons. The authors evaluated a lot of pre- and perioperative parameters, and the four risk factors identified by the multivariate analysis seem reasonable. Readers of the journal may have an interest in the results of this study, and the nomogram may help predict an incidence of postoperative ARDS in NSCLC patients who undergo pneumonectomy.
However, the result of the validation analysis might not have any meaning because the performance of the nomogram should be effective on the same data set. The performance of the nomogram might have been assessed on the different data sets.
- All abbreviations used in the manuscript should be explained at the first mention, such as BMI, ASA score, CCI, FEV1, and DLCO.
We added all explanations after the first mention in the text
- Postoperative ARDS should be defined in the Material and Methods section.
We added paragraph 2.6 (ARDS definition).
“ARDS was defined according to the 2012 Berlin definition:
- acute onset within 7 days after surgery with ventilation setting for positive end-expiratory pressure (PEEP) of ≥5 cmH2O and bilateral lung infiltration, detected through chest x-ray, cannot be fully explained by effusion, lobar, lung collapse, or nodules;
- absence of hydrostatic or cardiogenic pulmonary edema;
- partial pressure of arterial oxygen and fraction of inspired oxygen (PaO2/FiO2) 300 mmHg or less, and classified in 3 categories of severity: mild (200 mmHg < PaO2/FiO2<300 mmHg), moderate (100 mmHg < PaO2/FiO2<200 mmHg) and severe (PaO2/FiO2<100 mmHg).”
- When looking at the tables, some of the patients had missing data in many of the variables. How did you deal with such patients with missing data in the analyses?
There are missing data only concerning only: DLOC/DLCO-PPO, lung perfusion, intraoperative fluid balance and blood transfusions (respectively 19/12, 23 and one missing data out of 211 patients). This lack of data does not substantially affect the results.
The main lack of data concerns preoperative VO2 max, but this topic is widely specified in the text:
- materials and methods 2.2 paragraph: “In the last 4 years, our routine tests have included the cardio-pulmonary stress test for evaluating VO2max”
- results: “Vo2 max was recorded for only 53 patients (25.1%), 6 of whom developed ARDS. Based on these limited data, a significant increase of ARDS was observed with decreasing Vo2max level (Mantel-Haenszel P=0.03)”
- discussion: “Instead, we found a strong relationship between preoperative VO2 max, calculated by cardio-pulmonary stress test, and ARDS; however, we did not focus on this aspect and did not consider it in the multivariable analysis because this parameter was available only for 53 patients out of 211. Indeed, we only recently introduced the cardio-pulmonary stress test in our routine preoperative tests (since 2019); this factor will be furtherly investigated in the future”….. “We excluded the other significant parameter at multivariable analysis (intraoperative fluid balance) because of the need to preoperatively assess the risk of ARDS. Likewise, we did not include VO2max because of its unavailability for most patients”
- Even in the variables regarding vessels, the data of such variables seem unknown in some patients. Was not preoperative evaluation by CT of the thorax performed in every patient?
According to retrospective character of the study, we reviewed our single-center experience between 2014 and 2021. The lack of measurement in the first 6 patients of the series is due to the impossibility of evauation of preoperative CT-scan; indeed, preoperative images are removed approximately every 8 years from our systems.
5) As I mentioned above, the result of the validation analysis might not have any meaning because the performance of the nomogram should be effective on the same data set. The performance of the nomogram might have been assessed on the different data sets.
Re: Yes we agree with this reviewer, any nomogram requires validation on a different dataset, preferably external, before being used in the clinic. Unfortunately, we did not have access to data from independent series to perform ourselves a validation. We now mentioned that validation is required in the discussion. We also removed the sentence regarding the performance of the nomogram on the same data set (Overall, the total number of ARDS predicted by the nomogram (28.0) was similar to the observed number), which may be falsely interpreted as a validation.
- In the tables, all abbeeviations used should be explained, such as FEV1, DLCO, and BPCO.
We added explanations in the tables
- It is easier to understand if the columns of % are present in the patients with and without ARDS groups, respectively.
The percentage is indicated (in round brackets) next to each variable, referring both to the ARDS group and to the no ARDS group.
8) In the tables, the authors should present the number of patients with unknown data in every variable.
We added in each variables, the number for missing data.

Reviewer 2 Report
In the present paper Authors developed a nomogram to predict the incidence of post-operative ARDS after pneumonectomy for lung cancer. The study is well designed, aims and methods are clearly exposed. Among the study population of 211 patients, 28 (13.3%) developed post-operative ARDS. Overall 30-days mortality was 3.3%, 14.3% in patients with ARDS. The multivariable analysis found four main risk factors which can stratify patients into different risk classes for ARDS. I really appreciated authors’ efforts in this lacking and debated field of research. I only have minor revisions to suggest:
Materials and Methods section (page 3 line 95): although ASA, FEV1, DLCO, COPD and BMI are common abbreviations, easily found in papers focusing on lung resection outcomes, CCI is not. Please explain the first time it appears within the text. The same for HALP (line 121).
Result section (page 5 line 205): authors reported a median value of predicted probability of ARDS of 0.05%. It sounds to me a very low value. Could the authors check and confirm?
Discussion section (page 14 lines 352-360): authors state their classification allows a correct preoperative stratification of patients (I agree) in order to ensure the best care for these patients. Since the risk factors they found are not modifiable, the conclusion “in order to ensure the best care” is not supported, I guess the issue in this subset of high risk patients could be if proceed to surgery or propose alternative cancer treatments. The same for the sentence “the second step … is the correct surgical and anesthesiologist management”, I think specific an updated pneumonectomy programs are currently undergoing in all the certified thoracic surgery units, on the other hand the proposed nomogram could be useful for patients’ pre-operative counseling and selection. Please comment on this.
Conclusion section: the present paper shows some interesting findings in predicting ARDS after pneumonectomy, in my personal opinion a multicenter validation cohort is needed in order to assess how the nomogram works outside the calibration cohort. This should be mentioned as a future perspective.
Table 5: I see some differences in ARDS incidence, ICU and hospital stay but p values are lacking. Can they be reported in this table?
Author Response
REVIEWER 2
Comments and Suggestions for Authors
In the present paper Authors developed a nomogram to predict the incidence of post-operative ARDS after pneumonectomy for lung cancer. The study is well designed, aims and methods are clearly exposed. Among the study population of 211 patients, 28 (13.3%) developed post-operative ARDS. Overall 30-days mortality was 3.3%, 14.3% in patients with ARDS. The multivariable analysis found four main risk factors which can stratify patients into different risk classes for ARDS. I really appreciated authors’ efforts in this lacking and debated field of research. I only have minor revisions to suggest:
Materials and Methods section (page 3 line 95): although ASA, FEV1, DLCO, COPD and BMI are common abbreviations, easily found in papers focusing on lung resection outcomes, CCI is not. Please explain the first time it appears within the text. The same for HALP (line 121).
We added all explanations after the first mention in the text and in the tables.
Result section (page 5 line 205): authors reported a median value of predicted probability of ARDS of 0.05%. It sounds to me a very low value. Could the authors check and confirm?
The Median predicted probability was 0.05 OR 5%, (4.9% to be more precise). We corrected this in the text of the manuscript. (parapgraph 3.3)
Discussion section (page 14 lines 352-360): authors state their classification allows a correct preoperative stratification of patients (I agree) in order to ensure the best care for these patients. Since the risk factors they found are not modifiable, the conclusion “in order to ensure the best care” is not supported, I guess the issue in this subset of high risk patients could be if proceed to surgery or propose alternative cancer treatments. The same for the sentence “the second step … is the correct surgical and anesthesiologist management”, I think specific an updated pneumonectomy programs are currently undergoing in all the certified thoracic surgery units, on the other hand the proposed nomogram could be useful for patients’ pre-operative counseling and selection. Please comment on this.
I agree with the reviewer, but I think that my explanation was misinterpreted. The concept “to ensure the best care” was to be effectively considered in the context of choosing between surgery and a more conservative treatment such as radiotherapy and medical therapies; despite the the increased risks, if surgery remains the treatment of choice, “the second step” was an intensive and more attentive intra and post-operative management of these patients.
We changed some affirmations in the discussion paragraph:
“Our classification is easy, reliable and exclusively based on preoperative and easily available data; it allows a correct preoperative stratification of patients based on their functional (lung perfusion and DLCO<75%) and inflammatory (CRP level) status and their medical history (ASA score), in order to ensure the choice of the best treatment (between surgery and a more conservative treatment such as radiotherapy) for these patients and facilitate their global management. Thus, a correct interaction between thoracic surgeons, respiratory physicians, anesthesiologists, physiotherapists and dedicated ICU nurses is mandatory. The first step is the correct analysis of medical history and of lung function. The second step, especially for patients in the highest classes, if the treatment remains surgery, is the correct surgical and anesthesiologist management.”
Conclusion section: the present paper shows some interesting findings in predicting ARDS after pneumonectomy, in my personal opinion a multicenter validation cohort is needed in order to assess how the nomogram works outside the calibration cohort. This should be mentioned as a future perspective.
We add this concept in the conclusions paragraph.
Table 5: I see some differences in ARDS incidence, ICU and hospital stay but p values are lacking. Can they be reported in this table?
We now added p-values in the table

Round 2
Reviewer 1 Report
1) As the authors mentioned, the patients with missing data were excluded from some of the analyses. In DLCO/VA and Fluid balance, almost 10% of the patients were excluded because of missing data, and it might affect the results of the analyses. In the analyses, the data might be complemented by estimated values, such as the median of each variable. Otherwise, you should mention about the patients with missing data in the study limitation.
2) In the tables, the percentage may be separately presented in ARDS and no ARDS groups in each variable.
Author Response
1) As the authors mentioned, the patients with missing data were excluded from some of the analyses. In DLCO/VA and Fluid balance, almost 10% of the patients were excluded because of missing data, and it might affect the results of the analyses. In the analyses, the data might be complemented by estimated values, such as the median of each variable. Otherwise, you should mention about the patients with missing data in the study limitation.
Thanks for the remark.
We added an other phrases in the limitation's section.
"Data relating to some parameters such as DLCO and fluid balance were partially missing (about 10%); this aspect might have affected the results of the analyses"
2) In the tables, the percentage may be separately presented in ARDS and no ARDS groups in each variable.
Thanks for the remark. We corrected the tables by placing all the percentages in brackets, specifying at the top of the tables that the percentages were in the brackets. We added no other columns, in order to avoid making tables too complex and large.